# Polymethyl(1–Butyric acidyl)silane–Assisted Dispersion and Density Gradient Ultracentrifugation Separation of Single–Walled Carbon Nanotubes

**DOI:** 10.3390/nano12122094

**Published:** 2022-06-17

**Authors:** Hongming Liu, Qin Zhou, Yongfu Lian

**Affiliations:** Key Laboratory of Functional Inorganic Material Chemistry, Ministry of Education, School of Chemistry and Materials Science, Heilongjiang University, Harbin 150080, China; 2191007@s.hlju.edu.cn

**Keywords:** polymethyl(1–butyric acidyl)silane, dispersion, density gradient ultracentrifugation

## Abstract

Individual single–walled carbon nanotubes (SWNTs) with distinct electronic types are crucial for the fabrication of SWNTs–based electronic and magnetic devices. Herein, the water–soluble polymethyl(1–butyric acidyl)silane (BA–PMS) was synthesized via the hydrosilylation reaction between 3–butenoic acid and polymethylsilane catalyzed by 2,2′–azodibutyronitrile. As a new dispersant, BA–PMS displayed a quite good dispersing capacity to arc–discharged SWNTs and moderate selectivity for metallic species. The application of sucrose–DGU, the density gradient ultracentrifugation with sucrose as the gradient medium, to the co–surfactants (BA–PMS and sodium dodecyl sulfonate) individually dispersed SWNTs yielded metallic SWNTs of 85.6% purity and semiconducting SWNTs of 99% purity, respectively. This work paves a path to the DGU separation of the SWNTs dispersed by polymer–based dispersants with hydrophobic alkyl chains.

## 1. Introduction

In recent years, single–walled carbon nanotubes (SWNTs) have attracted much attention because of their excellent electrical, thermal, optical, and mechanical properties [1,2,3,4]. In particular, individual metallic SWNTs (m–SWNTs) and semiconducting SWNTs (sc–SWNTs) as promising materials in electrodes, logic circuits, field–effect transistors, and sensors have great potential applications [5,6,7,8,9,10,11,12]. To date, commercially available SWNTs mainly include chemical vapor deposition (CVD) yielded HiPco and CoMoCAT [8,11,13], laser–ablated [14], and arc–discharged ones [15], which invariably form large bundles composed of mixtures of 1/3 metallic and 2/3 semiconducting species [14]. In comparison with the others, arc–discharged SWNTs are longer, straighter, and more tightly bundled owing to their little defects, high aspect ratios, and strong van der Waals interactions, making it more difficult to disperse and isolate them in liquid media [15,16].

In recent years, the chirality indices (n,m)–controlled synthesis of SWNTs has been attracting more and more attention. Though the selective growth of sc–SWNTs or even SWNTs with a single chirality index has been realized via CVD on varying solid metal–containing catalysts [17], the stable and large–scale production of SWNTs with uniform chirality indices (n,m) remains a great challenge, far from meeting the demand of practical applications [18,19]. To date, post–treatment of the as–synthesized SWNTs is the most efficient and practical approach to obtaining individually dispersed metallic or semiconducting SWNTs. The as–synthesized SWNTs could be dispersed in some solvents by covalent or non–covalent chemical modifications [20,21], in which individual SWNTs effectively peeled from large bundles form stable dispersions. Quite different from covalent modifications, non–covalent modifications maintain the intrinsic electronic properties of SWNTs [20], and they are widely used to achieve individual metallic or semiconducting SWNTs. For non–covalent chemical modifications, the choice of dispersing agents is crucial to the effective individualization and efficient sorting of metallic or semiconducting SWNTs. Typical dispersing agents are limited to classic surfactants [22,23,24], ionic liquids [25], and biological substances [12,26,27] applicable in water media as well as varying polymers [28,29,30] applicable in water or non–water media. In the case of polymers, both the water–soluble nonaromatic species [31,32,33] and the non–water–soluble π–conjugated ones are selective for sc–SWNTs [34,35,36,37]. According to Chung et al., a series of nonaromatic σ–conjugated poly(dialkylsilane) are selective for semiconducting CoMoCAT SWNTs in THF [38]. Recently, we synthesized a water–soluble polymethylsilane derivative, polymethyl(1–undecylic acidyl)silane (Ua–PMS), by the hydrosilylation reaction between undecylenic acid and polymethylsilane, whose alkaline aqueous solution demonstrated a highly special selectivity towards metallic arc discharged SWNTs with large diameters and small chiral angles [39]. As far as we know, this is the first polymer derivative that displays selectivity for m–SWNTs in water media.

Even though some dispersing agents display a selective dispersion for metallic or semiconducting SWNTs to some extent, the achieved dispersions are still mixtures of them with various chiral indices. For effective sorting of individually dispersed SWNTs, researchers turn to gel chromatography [40,41], two–phase aqueous extraction [42,43], dielectrophoresis [44,45], and density gradient ultracentrifugation (DGU) [46,47]. Among these techniques, DGU has the advantages of easy operation, large–scale separation, and universal utility for various SWNTs over the others, making it the most popular technique to separate SWNTs in line with their diameter, electronic type, chirality, or even–handedness. In the DGU separation of SWNTs, a series of density gradients are constructed by gradient media. Because of its low viscosity and osmotic pressure, iodixanol easily forms self–generated gradients, which is beneficial for the highly efficient separation of SWNT [46]. However, the remaining iodine from iodixanol would degrade the electronic structures of SWNTs [47], and iodixanol is not easy to be removed from SWNTs. To overcome these problems encountered in iodixanol–DGU, Yanagi et al. developed sucrose–DGU, in which sucrose is inert to SWNTs and then easily isolated from SWNTs [47]. Additionally, sucrose is a much cheaper reagent than iodixanol. Apart from gradient media, the composition of SWNTs dispersion plays an important role in the successful DGU separation of SWNTs, in which the dispersed SWNTs in aqueous solutions of different dispersing agents generate a difference in buoyant density when subjected to high centripetal force. Though anionic surfactants [48,49,50,51] or DNA [46,47] assisted dispersion is largely applied to DGU separation, the SWNTs dispersed by two surfactants are necessary in some cases. Sodium dodecyl sulfate (SDS), for example, is indispensable for DGU separation of the sodium cholate (SC) [51], sodium deoxycholate (DOC) [50], or *N*-cocoyl sarcosinate [46] dispersed SWNTs in line with their electronic type. As for the polymers assisted dispersion of SWNTs, only those of nonionic block copolymers (Tetronic and Pluronic) have been utilized to separate metallic or semiconducting SWNTs in the aqueous iodixanol gradient media [52].

In this work, water–soluble polymethyl(1–butyric acidyl)silane (BA–PMS) was synthesized through the hydrosilylation reaction between polymethylsilane and 3–butenoic acid with 2,2′–azobisisobutyronitrile (AIBN) as the catalyst. Such obtained polysilane derivative demonstrates efficient dispersion of arc–discharged SWNTs and proper selectivity for m–SWNTs. When subjected to sucrose–DGU separation, the SWNTs dispersed in the aqueous solution of BA–PMS and co–surfactant SDS were successfully sorted to highly enriched m–SWNTs and sc–SWNTs.

## 2. Materials and Methods

### 2.1. Materials

SWNTs were prepared by arc discharge method with a homemade furnace. All of the following chemicals were of analytical grade and used without further purification: FeS (Beijing Yili Fine Chemicals Co, Ltd., Beijing, China), YNi_2_ (General Research Institute For Nonferrous Metals, Beijing, China), Ni powder (Sinopharm Chemical Reagent Co., Ltd., Shanghai, China), HCl (12 mol/L, Harbin Polytechnic Chemical Reagent Co., Harbin, China), toluene (Tianjin Kermel Chemical Reagent Co., Ltd., Tianjin, China), tetrahydrofuran (Tianjin Yongsheng Superfine Chemical Industry Co., Ltd., Tianjin, China), dichloromethylsilane (Shanghai Aladdin Biochemical Technology Co., Ltd., Shanghai, China), 3–butenoic acid (Shanghai Macklin Biochemical Co., Ltd., Shanghai, China), 2,2′–azobisisobutyronitrile (Shanghai Macklin Biochemical Co., Ltd., Shanghai, China), NaOH (Sinopharm Chemical Reagent Co., Ltd., Shanghai, China), DOC (Shanghai Aladdin Biochemical Technology Co., Ltd., Shanghai, China), SDS (Shanghai Aladdin Biochemical Technology Co., Ltd., Shanghai, China), sucrose (Shanghai Aladdin Biochemical Technology Co., Ltd., Shanghai, China).

### 2.2. Synthesis of Water–Soluble BA–PMS

As shown in Figure 1, water–soluble BA–PMS was synthesized by the following two–step reactions. The first step is the Wurtz dechlorination coupling reaction [53]. Sodium block (4.6 g) was stirringly dispersed into sodium sand in toluene (200 mL) at a temperature of 383 K. When cooling down to 343 K, dichloromethylsilane monomer (26 mL) was added dropwise under the protection of high–pure nitrogen. After refluxing for 8 h, the reaction mixture was filtered and the filtrate was evaporated with a rotary evaporator, thus obtaining the pale yellow and viscous polymethylsilane (PMS). The second step is the hydrosilylation reaction [54]. When AIBN (0.2 g) and PMS (2.2 g) were dissolved in tetrahydrofuran (THF) (40 mL), 3–butenoic acid (4.18 mL) was added dropwise at a temperature of 343 K. Under the protection of high–pure nitrogen, the reaction mixture was stirred for 10 h at a temperature of 343 K. Finally, the reaction mixture was evaporated with a rotary evaporator to remove THF solvent, and then the light yellow and viscous BA–PMS was obtained.

### 2.3. Preparation and Primary Purification of SWNTs

The arc–discharged preparation and primary purification of SWNTs were reported previously [39]. Briefly, the anode was a composite graphite rod doped with a powder mixture of Ni, YNi_2_, and FeS, and the cathode was a round graphite block with a radius of 20 mm. An arc was generated at a current of 100 A under a He atmosphere of 500 Torr. Usually, the anode was consumed for about 20 min and the as–prepared SWNTS were collected from the roof and wall of the arc discharge chamber. The as–prepared SWNTs were firstly heat–treated at 674 K for 2 h under an air atmosphere, and then the residues were ground with a ball mill for 10 min. After the fine powders were soaked in concentrated HCl for 24 h, the mixture was filtrated and washed with a large amount of water until the pH value of the filtrate is around 7, then obtained the primarily purified SWNTs. The above purification procedure was followed by SEM observation, TG analyses, Raman scattering, and optical absorption spectroscopies (see Appendix A).

### 2.4. The Dispersion Procedure of the Primarily Purified SWNTs

In a typical dispersion experiment, the above synthesized BA–PMS was dissolved in 100 mL of deionized water. After the pH value of the solution was adjusted to 9 with 0.5 mol/L NaOH, a clear and transparent aqueous solution of BA–PMS was achieved. The primarily purified SWNTs (15 mg) were dispersed in 50 mL alkaline aqueous solution of BA–PMS under discontinuous sonication (Vibra–cell of Sonics VCX 750, Newtown, CT, USA) for 8 h at a temperature of 283 K. The resulting dispersion was centrifuged (Hitachi CP70 MX, P70 AT2 angular rotor, Tokyo, Japan) at a rotation speed 15,000 rpm for 2 h, and the upper 80% supernatant, i.e., BA–PMS–dispersed SWNTs, was carefully collected.

### 2.5. Separation of BA–PMS–Dispersed SWNTs by Sucrose–DGU

A series of aqueous solutions of sucrose with mass fractions of 30 wt%, 40 wt%, 50 wt%, and 60 wt% were prepared, respectively, which were applied to construct density gradients. The density gradient column was formed by stacking a series of layers in a 10 mL polyethylene centrifuge tube with the following sucrose solutions: 60 wt% (2 mL), 50 wt% (2 mL), 40 wt% (2 mL), and 30 wt% (2 mL). When the co–surfactant SDS (0.1 g) was dissolved in the BA–PMS–dispersed SWNTs (10 mL), the SWNTs dispersion applicable for DGU is ready. After 2 mL of the SWNT dispersion was added to the 30 wt% layer, the polyethylene centrifuge tube was sealed with a capping machine and then subjected to ultracentrifugation (Hitachi CP70 MX, P65 AT3 vertical rotor) at a rotation speed of 25,000 rpm for 20 h at a temperature of 20 °C. All of the fractions were successively collected using a long needle from the top of the centrifuge tube.

### 2.6. Characterization Methods

Visible near–infrared (Vis–NIR) optical absorption spectra were recorded by a dual–beam spectrometer (UV3600, Shimadzu, Shanghai, China) with a spectral resolution of 0.2 nm in a 10 mm quartz cell; Fourier transform infrared (FT–IR) spectra were measured by a Fourier transform infrared spectrometer (Spectrum one, PerkinElmer, Shelton, CT, USA) in transmission mode, and samples were prepared by dropping the solution of 3–butenoic acid, PMS and BA–PMS on the surface of a pressed KBr sheet with a capillary tube, respectively; Scanning electron microscope (SEM) was tested on a Sigma 500 (Carl ZEISS, Jena, Germany); Thermogravimetric Analysis (TG) was texted on an SDT2960 (TA, New Castle, DE, USA); Transmission electron microscope (TEM) and atomic force microscope (AFM) observations were conducted by a JEM–2100 (JEOL, Tokyo, Japan) at an accelerator voltage of 200 kV and an AFM/SPM system (SPM 5100, Agilent, Palo Alto, CA, USA) in tap mode, respectively. Samples were fabricated by filtering the centrifugates of the BA–PMS dispersed SWNTs with a 0.1 μm PTFE membrane. After successive washing with a proper amount of methanol and acetone, the residue on the membrane was ultrasonically dispersed in ethanol. The ethanol dispersions of SWNTs were dropped on a 200 mesh Cu grid or a clean silicon substrate for TEM or AFM observation. Resonance Raman spectra were recorded by a microlaser Raman spectrometer (HR800, Jobin Yvon, Paris, France) under excitation with laser wavelengths of 458, 514, 532, and 633 nm, respectively. Samples were prepared by filtering the centrifugates of the BA–PMS dispersed SWNTs with a 0.1 μm PTFE membrane, followed by vacuum drying for 1 h.

## 3. Results

### 3.1. The Synthesis of BA–PMS

FT–IR spectra were applied to analyze and monitor the synthesis of BA–PMS. Figure 1 shows the FT–IR spectra of PMS, 3–butenoic acid, and BA–PMS, respectively. The absorption peak at 2100 cm^−1^ in Figure 1a is attributed to the stretching vibration of the Si–H bond, which is a characteristic peak of PMS. The absorption peak at 1640 cm^−1^ in Figure 1b is identified as the stretching vibration of the C=C bond, a characteristic of 3–butenoic acid. It is obvious that the characteristic Si–H absorption in BA–PMS is significantly weakened in comparison with that observed in PMS, and the characteristic C=C absorption in BA–PMS almost completely disappears in comparison with that in 3–butenoic acid. These evolvements in the FT–IR vibrational spectral features confirm that a hydrosilylation reaction occurred between the largely broken Si–H bond of PMS and the C=C bond of 3–butenoic acid, i.e., 1–butyric acidyl is successfully introduced to the side chain of PMS via an Anti–Marshall addition reaction, leading BA–PMS to be soluble in water. Moreover, as shown in Appendix A the water–soluble BA–PMS has no absorbance in the Vis–NIR region, making it an applicable dispersing agent for SWNTs owing to no interference with the detection of metallic or semiconducting nanotubes.

### 3.2. The Individual Dispersion of Arc–Discharged SWNTs with BA–PMS

To investigate the long–standing stability of the BA–PMS dispersed SWNTs, we dispersed arc–discharged SWNTs in water, the aqueous solutions of BA–PMS and DOC, respectively, by a tip sonicator. It can be seen from Appendix A that both BA–PMS and DOC–assisted dispersions of SWNTs are homogeneous and transparent solutions, which still keep stable even after standing for 90 days. In contrast, the SWNTs dispersed in water precipitate immediately once the ultrasound treatment stops. Moreover, the color of the BA–PMS–assisted dispersions of SWNTs is deeper than that of the DOC–assisted ones. Therefore, it is concluded that BA–PMS is superior to DOC for the stable dispersion of SWNTs in water–soluble solutions.

TEM and AFM observations are directive and effective for the visual characterization of individually dispersed SWNTs. Displayed in Figure 2 are the typical AFM and TEM images of SWNTs dispersed in an aqueous solution of BA–PMS. In Figure 2a, we observe many individually dispersed SWNTs. Even though some residues of BA–PMS are observable and the walls of SWNTs are quite clean, indicating that most of the dispersant (BA–PMS) was removed by successive washing with methanol and acetone. Moreover, the diameter of an individual SWNT (d_t_) was measured to be 1.4 nm, which is consistent with the average diameter of arc–discharged SWNTs. From Figure 2b, we observed a curved filamentous structure longer than 3 μm with many dome–like structures. The filamentous structure is attributed to an individual carbon nanotube wrapped with varying quantities of BA–PMS molecules, and those dome–like structures on the substrate come from the aggregates of BA–PMS molecules, which are composed of three distinct parts, i.e., a horizontally self–assembled monolayer (0.8 nm), a transition state (2 nm) and vertical alignment (8.0 nm) [55,56]. In comparison with those observed previously for the complex of Ua–PMS and SWNTs [39], the dome–like structures with large sizes reduce drastically, also indicative of the efficient removal of BA–PMS by successive washing with methanol and acetone. According to the height analysis results in Figure 2b, the diameter of the filamentous structure is around 2.4 nm. Since the sample for AFM observation was made by dropping the ethanol BA–PMS dispersion of SWNTs onto a clean silicon substrate, the well–dispersed SWNTs would inevitably be gathered to some extent and the BA–PMS molecules would aggregate to form the matrix observed in Figure 2b after the evaporation of ethanol solvent. Considering the thickness and orientation of BA–PMS molecules covered on the surface of carbon nanotubes, it is reasonable for us to describe the SWNTs observed here as individuals, which is in agreement with the results of TEM observation.

Vis–NIR optical absorption spectroscopy is highly effective to detect the dispersion state as well as the conductivity type of SWNTs. Shown in Figure 3a are the Vis–NIR absorption spectra of the primarily purified and the BA–PMS dispersed SWNTs. For the arc–discharged SWNTs, the bands in 600–800 nm, 800–1200 nm, and 400–600 nm regions are usually attributed to the first van Hove transition of m–SWNTs (M_11_), the second van Hove transition of sc–SWNTs (S_22_), and the overlapped third and fourth van Hove transitions of sc–SWNTs (S_33_ + S_44_), respectively. In sharp contrast to that of the primarily purified SWNTs, the Vis–NIR optical absorption spectrum of the BA–PMS dispersed SWNTs is obviously blue–shifted, enriched in many fine absorption peaks, inverted for the relative absorbance of S_22_ band to M_11_ band, and decreased in background absorption, evidencing that the bundles of SWNTs are successfully unbundled and individualized in the aqueous BA–PMS dispersion of SWNTs.

The relative abundance of metallic to semiconducting SWNTs in the aqueous BA–PMS dispersion of SWNTs was estimated by the method proposed by Haddon et al. [57]. Shown in Figure 3b–e are the absorption curves of the primarily purified and BA–PMS dispersed SWNTs in the M_11_ and S_22_ regions, respectively. In each of them, a linear baseline is adopted to partition the total integral area under the curve, AA(T), into the integral area under the linear baseline, AA(I), and the integral area between the curve and the linear baseline, AA(M) or AA(S). Under the circumstances, the relative purity of metallic and semiconducting tubes, i.e., RP(M) and RP(S), could be estimated by AA(M)/AA(T) and AA(S)/AA(T), respectively. In line with Figure 3b,c, RP(M) and RP(S) were derived to be 0.0130 and 0.0677, respectively. Thus, the relative content of metallic to semiconducting SWNTs for the primarily purified SWNTs is about 0.1920 (0.0130/0.0677). Similarly, based on Figure 3d,e, RP(M) and RP(S) were derived to be 0.0596 and 0.2111, respectively, thus the relative content of metallic to semiconducting SWNTs for the BA–PMS–dispersed SWNTs is about 0.2823 (0.0596/0.2111). In this regard, it means that the relative content of metallic SWNTs in the BA–PMS–dispersed SWNTs is about 1.47 (0.2823/0.1920) times that in primarily purified SWNTs, indicating that BA–PMS is selective for the dispersion of m–SWNTs.

Raman spectroscopy plays an essential role in the characterization of carbon materials. Especially, the radial breathing mode (RBM) in the range of 100–300 cm^−1^ and the tangential Raman mode (G band) near 1590 cm^−1^ are the fingerprints of SWNTs. Shown in Figure 4 are the Raman spectra of the primarily purified and the BA–PMS dispersed SWNTs. In the RBM range, we note that all of the RBM peaks of the BA–PMS dispersed SWNTs are systematically blue–shifted in comparison with those of the primarily purified ones, regardless of their relative intensities. For example, the peaks at 163 and 149 cm^−1^ for the primarily purified SWNTs are blue–shifted to 175 and 166 cm^−1^ for the BA–PMS dispersed SWNTs in the 458 and 633 nm excitation spectra, respectively. We observed previously a similar blue shift in the RBM frequency for the well–dispersed SWNTs relative to those before dispersion [39,40], which is a direct result of unbundling–induced blue shift for the electronic transition, making it possible for SWNTs with different diameters in a bundle or individual states to resonate with the same laser excitation. On the other hand, because of the curvature of the rolled–up graphite sheet, the G band of SWNTs splits to the higher frequency component (G^+^) attributed to vibrations along the direction of the nanotube axis and the lower frequency component (G^−^) associated with vibrations along the circumferential direction. It can be observed from Figure 4 that the upper sharp G^+^ bands narrow down when the primarily purified SWNTs are dispersed in the aqueous solution of BA–PMS. Both the blue–shifted RBM band and the narrowed G^+^ band observed in Figure 4 are ascribed to the efficient unbundling and effective individualization of the BA–PMS dispersed SWNTs. Moreover, for the arc–discharged SWNTs, metallic and semiconducting species mainly resonate with 458 and 633 nm laser excitations, respectively. In the 458 nm excitation spectrum, the relative integral intensity of G^−^ to G^+^ bands for the BA–PMS–dispersed SWNTs is nearly the same as that for the primarily purified SWNTs, which is a character of sc–SWNTs (Appendix A). In contrast, in the 633 nm excitation spectrum, the G^−^ band of the primarily purified SWNTs is fitted to a peak, and that of the BA–PMS dispersed SWNTs to two peaks. The relative integral intensity of G^−^ to G^+^ bands is estimated to be 4.58 for the BA–PMS–dispersed SWNTs, while it is about 2.66 for the primarily purified SWNTs (Appendix A). In line with Akasaka et al. [58], the enhancement in the lower frequency component G^−^ band is a result of the enrichment of m–SWNTs. Therefore, the relative content of m–SWNTs in the BA–PMS–dispersed SWNTs is estimated to be 1.72 (4.58/2.66) times that in the primarily purified SWNTs, which is consistent with the above result extracted from their optical absorption spectra, offering another support for the selective dispersion of BA–PMS towards m–SWNTs.

### 3.3. DGU Separation of the BA–PMS–Dispersed SWNTs

With the success in the individualization of arc–discharged SWNTs, DGU separations were conducted on the surfactant BA–PMS and co–surfactant SDS dispersed SWNTs in density gradients generated with sucrose solutions. Other than Pluronic block copolymers in iodixanol–DGU [52], it is unnecessary to load surfactant BA–PMS throughout the density gradient. In sharp contrast to the dynamic interaction between Pluronic and SWNTs, the chain–like BA–PMS macromolecules wrapping on SWNTs are difficult to detach from the sidewalls of carbon nanotubes.

Figure 5a presents a photograph of the centrifuge tube after DGU, from which two bands roughly separated by over 1 cm are observed in the centrifuge tube. The upper band successively consisted of layers with green, brown, and red–brown colors down the tube. After fractionation layer by layer, the top green and bottom red–brown layers as displayed in Figure 5c were evidenced to be highly enriched in m–SWNTs and sc–SWNTs, respectively. As shown in Figure 5b, in comparison with that before centrifugation, the absorbance of metallic species is largely increased for SWNTs in the top green layer, and that of semiconducting ones is significantly enhanced for SWNTs in the bottom red–brown layer. Correspondingly, the absorbance of semiconducting species is decreased greatly for SWNTs in the top green layer, and that of metallic ones has almost completely vanished for SWNTs in the bottom red–brown layer. Moreover, in comparison with the absorption curve before ultracentrifugation, the M_11_ band of the SWNTs in the top green layer extends about 100 nm in the shorter wavelength end, while the S_33_ band of the SWNTs in the bottom red–brown layer also extends about the same extent in the longer wavelength end. Considering the large distribution in the diameters for arc–discharged SWNTs [15], we ascribe these extensions to be owing to the m–SWNTs with smaller diameters and sc–SWNTs with larger diameters, respectively. Because of the overlap in the transitions of M_11_ and S_33_ bands, it is only possible to distinguish them in highly enriched metallic or semiconducting SWNTs. As far as we are acknowledged, the overlap in the S_33_ and M_11_ optical absorption bands has never been reported previously for the arc–discharged SWNTs, even though Hersam et al. [51] found a similar overlap for the laser–ablation–grown SWNTs separated in a co–surfactant solution (SDS/SC). In addition, the absorption curve of the SWNTs in the top green layer is red–shifted to some extent relative to that of the BA–PMS–dispersed SWNTs, indicative of the enrichment of SWNTs of smaller diameters.

According to Hersam et al. [52], the enrichment in sc–SWNTs could be calculated as a percentage of the baseline–corrected area of the S_22_ peak divided by the sum of the baseline–corrected S_22_ and M_11_ peak areas. In the same way, the enrichment in m–SWNTs could be calculated as a percentage of the baseline–corrected area of the M_11_ peak divided by the sum of the baseline–corrected S_22_ and M_11_ peak areas. It can be seen from Appendix A that the estimated purity of the m–SWNTs in the top green layer is 83.6%, which is seven times (83.6%/11.3%) of that in the primarily purified SWNTs (see Appendix A). As a matter of fact, the peak absorbance (A) correlates with the concentration of SWNTs (c in mg/mL) through Beers law (A = εlc), where ε is the extinction coefficient (mL/mg.cm) and l is the path length (1 cm in this work). Considering the data reported previously [26], i.e., ε is 52.04 and 60.59 mL/mg.cm for m– and sc–SWNTs, respectively, the purity of m–SWNTs in the top green layer is estimated to be 85.6%. As for the s–SWNT in the bottom red–brown layer, it is impossible to estimate their purity by the same method, since the effective M_11_ transition from the m–SWNTs is not visible and therefore cannot be measured owing to the low sensitivity of absorption spectroscopy and error propagation. Alternatively, we adopted the procedure proposed by Ding et al. To estimate the purity of sc–SWNTs in bottom red–brown layer [33]. Therefore, the brown curve in Figure 5b was re–plotted with absorbance vs. energy expressed as wavenumber in Appendix A, from which absorption peak ratio φ_i_ = A_CNT_/(A_CNT_ + A_B_) is defined, where A_CNT_ was the enveloping area of the M_11_ and S_22_ bands enclosed by the linear baseline in the region from 8500 to 15,139 cm^−1^, attributing to the nanotubes, and A_B_ was the area covered by the linear baseline of the same region, attributing mostly to the π–plasmon of carbonaceous impurities. The integration of the absorption curve of Appendix A leads to a value of 0.454, which is higher than the φ_i_ ratios for the Ding’s and Blackburn’s samples (0.403 and 0.387, respectively) [33,59], indicating an equivalent or higher sc–SWNTs purity of our sample (≥99%). We bet this is a direct result that the removal of the overlap between the S_22_ and M_11_ absorption features leads to a lower background intensity (i.e., a small A_B_ value) in this region. Moreover, based on the data shown in Appendix A, the recovery yields of m– and sc–SWNTs were estimated to be 4.6% and 1.8%, respectively. Additionally, from the performance features of DGU separation of SWNTs listed in Appendix A, it can be seen that this work achieved m–SWNTs with comparable or even higher purity than some DGU separations and sc–SWNTS with the highest purity. It should be pointed out that other than those extensively applied iodoxanol–DGU, sucrose–DGU offers an environment friendly and economic approach to effectively separate m– and sc–SWNTs [46,51,52,60,61,62,63]. On the other hand, this work opens the way for the DGU separation of m– and sc–SWNTs dispersed by chain–like polymers.

Shown in Figure 6 are the Raman spectra of the SWNTs in the top green layer and the bottom red–brown layer, respectively. In the RBM range, it is obvious that the peak assignable to sc–SWNTs in the bottom red–brown layer is much stronger than that in the top green layer for the 458 (Figure 6a), 514, or 532 nm excitation spectrum (Appendix A), whereas the peak assignable to m–SWNTs in the top green layer is significantly enhanced relative to that in the bottom red–brown layer for the 633 nm excitation spectrum (Figure 6b). On the other hand, in the 458, 514, and 532 nm (Appendix A) excitation spectra, the experimental shape of the G band can be fitted by two highly symmetric Lorentzian functions, in which the G^−^ peak is much narrower than the G^+^ peak, indicating that sc–SWNTs exist both in the top green layer and in the bottom red–brown layer. Valuable information can be extracted from the 633–nanometer excitation spectra (Appendix A), in which the experimental shape of the G band for the SWNTs in the top green layer can be fitted by two Lorentzian functions and a highly asymmetric Breit–Wigner–Fano (BWF) line shape. In contrast, that for the SWNTs in the bottom red–brown layer displays the same feature as that observed in the 458, 514, or 532 nm excitation spectrum. Thus, it is concluded that m–SWNTs and sc–SWNTs are highly enriched in the top green layer and the bottom red–brown layer, respectively, i.e., the separation of SWNTs in line with electronic types was realized by chain–like BA–PMS assistant dispersion and following density gradient ultracentrifugation.

### 3.4. Assignment of the Chiral Indices (n,m) for the BA–PMS–Dispersed SWNTs before and after DGU

Shown in Figure 7 are the RBM Raman features of BA–PMS–dispersed SWNTs under various laser excitations. In the 458 nm excitation spectrum (Figure 7a), the RBM peaks in the range of 130–150 cm^−1^ belong to m–SWNTs, while those in the range of 150–212 cm^−1^ to sc–SWNTs. The strong peak at the wavenumber of 172 cm^−1^ could be assigned to the semiconducting (14,6) SWNT with a diameter of 1.39 nm, whereas the shoulder peaks at 147, 159, 188, and 197 cm^−1^ are assigned to the metallic (19,4) SWNT with a diameter of 1.66 nm and semiconducting (16,6), (17,0) and (13,5) SWNTs with diameters of 1.54, 1.33 and 1.26 nm, respectively. Similarly, semiconducting (18,1), (14,6), (15,2), and (13,6) SWNTs with diameters of 1.45, 1.39, 1.26, and 1.32 nm are respectively detected in the 514 and 532 nm excitation spectra (Appendix A). Since the excitations of 458, 514, and 532 nm mainly resonate with semiconducting species for arc–discharged SWNTs, the above results do not mean that m–SWNTs are absent from the BA–PMS dispersed SWNTs. In the 633 nm excitation spectrum (Figure 7b), the peaks in the range of 110–166 cm^−1^ are attributed to sc–SWNTs, while the peaks in the range of 166–224 cm^−1^ to m–SWNTs. The RBM peaks at 149 and 166 cm^−1^ are identified as semiconducting (15,10) and (16,6) SWNTs with diameters of 1.71 nm and 1.54 nm, respectively, whereas the RBM peaks at 177, 191, and 210 cm^−1^ are identified as metallic (15,3), (14,5), and (14,2) SWNTs with diameters of 1.34, 1.31, and 1.18 nm, respectively.

On the other hand, according to the electronic energy band theory, the fine features observed in the optical absorption spectra of SWNTs are applicable to the assignment of their (n, m) chiral indices [57]. As shown in Appendix A, metallic (9,9), (15,3), (14,5), (13,7), (18,0) and (15,6) species as well as semiconducting (12,4), (11,6), (10,8), (14,3), (13,5), (17,0), (14,6), (16,5) and (16,6) ones are assigned for the SWNTs dispersed in the aqueous solution of BA–PMS. It is obvious that all of the chiral indices assigned in line with the principal fitted RBM peaks in Figure 7 are agreement quite well with the results of the fine optical features.

To identify the chiral indices of SWNTs after DGU separation, resonant Raman scattering spectra were also collected for the SWNTs both in the top green layer and in the bottom brown layer at the excitations of 458, 514, 532, and 633 nm, respectively. In the 458 nm excitation spectra (Figure 8a), the RBM peaks originating from the SWNTs in the bottom brown layer are assigned to semiconducting (13,9), (14,6), (17,0), and (13,5) tubes (Appendix A), and those from the SWNTs in the top green layer to semiconducting (16,6), (14,6) and (13,5) tubes along with a metallic (13,13) one (Appendix A). In the 633–nanometer excitation spectra (Figure 8b), the RBM peaks originated from the SWNTs in the top green layer are assigned to metallic (18,0), (14,5), (15,3), (12,6), (9,9) and (15,0) tubes along with two semiconducting (13,11) and (16,6) ones (Appendix A), and those from the SWNTs in the bottom brown layer to metallic (15,3) and (9,9) tubes along with semiconducting (19,5) and (15,8) ones (Appendix A). Even though sc–SWNTs are detected in the top green layer, their intensities are much smaller than their metallic counterparts. On the opposite, the normalized Raman intensities of the m–SWNTs in the bottom brown layer are also much smaller than those in the top green layer. Additionally, in the 514 and 532 nm excitation spectra, semiconducting (13,6), (14,6), and (15,2) tubes (Appendix A) are also detected for the SWNTs in the bottom brown layer or the top green layer.

Among the BA–PMS dispersed SWNTs assigned by Raman spectroscopy, metallic (14,2) and semiconducting (15,10) ones have the minimum (1.182 nm) and maximum (1.706 nm) diameters, which correspond to the optical absorptions at 584 and 633 nm, respectively. Similarly, the metallic (15,0) tube in the top green layer and the semiconducting (19,5) in the bottom brown layer are of the minimum (1.174 nm) and the maximum (1.717 nm) diameters after DGU separation, which corresponds to the optical absorptions at 568 and 637 nm, respectively. Thus, the Raman characterization results shown above also confirm the overlap between M_11_ and S_33_ transition bands for arc–discharged SWNTs, which is consistent with that extracted from the optical absorption spectra in Figure 5b (The chiral information of all SWNTs designated in the Raman spectra is summarized in Appendix A).

### 3.5. DGU Separation Mechanism

Aiming at the separation of SWNTs with different electronic types, a cosurfactant mixture was applied to achieve a change in the buoyant densities between m– and sc–SWNTs [26,51,64,65]. As shown in Figure 2, the effective individualization of arc–discharged SWNTs was realized by ultrasonic dispersion in the aqueous solution of a branched linear chain surfactant BA–PMS molecules. After the addition of the second linear chain surfactant SDS to the centrifugate after pre–ultracentrifugation occurs a change in the buoyant density between m– and sc–SWNTs is induced by the various numbers of adsorbed SDS molecules. The adsorption of SDS molecules might take place by desorption of BA–PMS molecules from walls of SWNT with subsequent adsorption of SDS. Since the CH–π interaction between SDS and SWNTs is stronger than that between BA–PMS and SWNTs, the adsorbed BA–PMS molecules are expected to be easily replaced by SDS. Considering the lower molar volume of SDS than BA–PMS and the larger water cluster around the hydrated sulfonic acid groups than that around the hydrated carboxyl groups [26,51,64,65], the replacement of BA–PMS by SDS would result in a decrease in the buoyant density of the tube–surfactant complex. On the other hand, quite different from sc–SWNTs there are some free electrons in the Fermi level of m–SWNTs, and the CH–π interactions between SDS and m–SWNTs must be stronger than that between SDS and sc–SWNTs. Thus, it is concluded that there are more SDS molecules adsorbed on m–SWNTs than on sc–SWNTs through the replacement of BA–PMS, which is responsible for the enrichment of m– and sc–SWNTs in the top green and the bottom brown layers, respectively.

The above DGU separation was also performed with the Ua–PMS dispersed SWNTs, from which m–SWNTs are expected to be easily extracted because of their higher selectivity for m–SWNTs than BA–PMS. However, it is observed that all SWNTs were positioned in the lower section of the centrifuge tube with no separation of SWNTs (see Appendix A). Since the length of the alkyl chain in SDS is comparable with that in Ua–PMS, the change in their CH–π interactions with SWNTs is not large enough to support the efficient replacement of Ua–PMS by SDS. Therefore, the different DGU behaviors for the surfactant polysilane derivatives dispersed SWNTs evidenced that the effective replacement of the adsorbed polysilane derivatives is crucial for the efficient separation of m– and sc–SWNTs.

## 4. Conclusions

Water–soluble BA–PMS was evidenced to be an efficient dispersant for the arc–discharged SWNTs, showing a moderate selectivity for metallic tubes. With environment–friendly and inexpensive sucrose as gradient medium, sucrose–DGU was established and applied to the separation of the BA–PMS dispersed SWNTs. With the assistance of co–surfactant sodium dodecyl sulfonate, highly pure m–SWNTs and sc–SWNTs were collected, respectively. Optical absorption and Raman scattering spectroscopies confirmed that the transition of the M_11_ band overlaps with that of the S_33_ band for the highly enriched arc–discharged SWNTs. In addition, more (n, m) chiral indices were assigned for the BA–PMS–dispersed SWNTs after rather than before DGU separation. In the aqueous solution of co–surfactant BA–PMS and SDS, the difference in the buoyant densities of m– and sc–SWNTs is ascribed to the competitive CH–π interactions of BA–PMS and SDS with m–SWNTs. With the successful DGU separation of the m– and sc–SWNTs dispersed by polymer–based dispersants with hydrophobic alkyl chains, it is highly possible to separate SWNTs in line with their diameter, and chirality and handedness by DGU treatment of the same kind of dispersions.

## Data Availability

The data presented in this study are available on request from the corresponding author.

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
