# Peer review of "Polymethyl(1–Butyric acidyl)silane–Assisted Dispersion and Density Gradient Ultracentrifugation Separation of Single–Walled Carbon Nanotubes"

_nanomaterials, 2022, doi:10.3390/nano12122094_

Round 1
Reviewer 1 Report
The manuscript by Liu et al describes the synthesis and purification of single walled nanotubes. On overall, I appreciated the manuscript, which is reasonably well structured. Methodology is described in sufficient details, and the description of the purification method is certainly interesting. With a couple of minor caveats, I would recommend publication in nanomaterials.
1) my most severe concern is related to English. There a some sentences which are mis-forumated, such as:
- row 200: "are homogeneous and transparent properties" should be "have homogeneous..." or maybe "are.... solutions"
-row 207-209: english should be checked
- row 313, 327, 330: check the English
These are only selected examples, the authors should check carefully their manuscript.
2) While the scientific part is sound, I wonder which is the recovery yield of the purified nanotubes. It would be great if the authors were able to provide a recovery, in addition to purity. Could the authors provide some insight in this sense?
Reviewer 2 Report
See attached doc
